# Hijacked Immune Cells in the Tumor Microenvironment: Molecular Mechanisms of Immunosuppression and Cues to Improve T Cell-Based Immunotherapy of Solid Tumors

**DOI:** 10.3390/ijms22115736

**Published:** 2021-05-27

**Authors:** Emre Balta, Guido H. Wabnitz, Yvonne Samstag

**Affiliations:** Section Molecular Immunology, Institute of Immunology, Heidelberg University Hospital, 69120 Heidelberg, Germany; guido.wabnitz@immu.uni-heidelberg.de

**Keywords:** TME, immunosuppression, cancer immunotherapy, CAR T cells, TILs, ROS

## Abstract

The understanding of the tumor microenvironment (TME) has been expanding in recent years in the context of interactions among different cell types, through direct cell–cell communication as well as through soluble factors. It has become evident that the development of a successful antitumor response depends on several TME factors. In this context, the number, type, and subsets of immune cells, as well as the functionality, memory, and exhaustion state of leukocytes are key factors of the TME. Both the presence and functionality of immune cells, in particular T cells, are regulated by cellular and soluble factors of the TME. In this regard, one fundamental reason for failure of antitumor responses is hijacked immune cells, which contribute to the immunosuppressive TME in multiple ways. Specifically, reactive oxygen species (ROS), metabolites, and anti-inflammatory cytokines have central roles in generating an immunosuppressive TME. In this review, we focused on recent developments in the immune cell constituents of the TME, and the micromilieu control of antitumor responses. Furthermore, we highlighted the current challenges of T cell-based immunotherapies and potential future strategies to consider for strengthening their effectiveness.

## 1. Composition and Heterogeneity of the Tumor Microenvironment

During the multi-stage development of tumors, normal cells acquire characteristics of cancer cells which have been postulated as continuous proliferative signaling, evasion of growth suppressors, resistance to cell death, immortality, induction of angiogenesis, and activation of invasion and metastasis [1]. In addition to these well-accepted postulates of cancer hallmarks, avoiding immune destruction and deregulating cellular energetics are described as emerging hallmarks [2]. By exhibiting these features, a high genetic diversity within the tumor arises. In an established tumor, the heterogeneity in the cancer cell population can largely be explained by the selective advantage of certain subclones that outgrow the other clones in the tumor environment.

Furthermore, tumors are not only masses of a heterogeneous population of cells with neoplastic transformation, but also contain non-transformed immune and non-immune cells [3]. This dynamic network of cells and macromolecules, which acquires inherent complexity during cancer progression or upon therapeutic intervention, forms the tumor microenvironment (TME). The non-immune cell infiltrates are composed of cancer-associated fibroblasts (CAFs), blood, and lymphatic vasculature cells. The immune cell composition is variable in different tumors, but generally includes quantitatively and functionally different populations of CD4+ T cells, CD8+ T cells, natural killer (NK) cells, dendritic cells (DCs), tumor-associated macrophages (TAMs), tumor-associated neutrophils (TANs), myeloid-derived suppressor cells (MDSCs), and B cells (Figure 1). The non-cellular components of the TME are the extracellular matrix (ECM) and soluble factors.

Understanding the composition and function of immune cell infiltrates of the TME is important for both prognosis and for designing optimal treatment modalities. This can be best understood from the failure of conventional therapies only aiming to directly target tumor cells without considering the TME. In recent years, characterization of the TME as cold (T cell non-inflamed) or hot (T cell inflamed) has already contributed significantly to successful therapies [4]. Immunologically hot tumors have higher T cell infiltrates and respond to immunotherapies such as checkpoint blockade inhibition [5]. Conversely, immunologically cold tumors, such as glioblastoma and pancreatic cancers, are resistant to checkpoint blockade therapies [6,7]. The immunosuppressive environment that leads to the exclusion of immune cells can be better defined in cold tumors. For such tumors, targeting of specific cell types such as TAMs can be combined with T cell-based immunotherapies. Indeed, first examples for this strategy already exist [5].

## 2. Tumor Immune Surveillance and Immunoediting

The concept of immunosurveillance has long been debated [8,9] since the first hypothesis of immunosurveillance was formulated by Paul Ehrlich in 1909 [10]. According to this theory, tumor cells are eradicated by our immune cells before they are clinically manifested. Even though this concept significantly contributed to research and understanding of antitumor immunity, it only explains the first step of cancer progression, namely elimination by immune cells. Later, the concept of cancer immunoediting was developed [11]. According to this concept, during tumor growth, immunoediting occurs in three phases, namely elimination, equilibrium, and escape. In the elimination phase, immune cells recognize and eliminate nascent tumor lesions that have developed because of failure of intrinsic tumor suppressor mechanisms. However, when the tumor cells are not completely cleared, this partial tumor cell elimination leads to an equilibration phase. In this phase, tumor outgrowth is controlled by the immune cells, but tumor cells continue to evolve, accumulating further mutations. Thus, further clones of tumor cells with different genetic modifications are generated. These new clones of tumor cells can resist, avoid, and suppress the antitumor immune response. Subsequently, they enter the escape phase, during which progressive tumor outgrowth takes place, which results in clinical manifestation of the tumor [12,13].

It should be noted that the escape phase is not only the result of tumor-intrinsic modifications, but is also evoked by the development of an immunosuppressive TME over time. Importantly, an immunosuppressive TME contains hijacked immune cells which downmodulate the T cell-controlled antitumor immune responses. Thus, understanding the contribution of hijacked immune cells to immunosuppression and the therapeutic strategies to overcome these obstacles have been the major focus in recent decades. Several mechanistic insights into the contribution of hijacked immune cells to immunosuppression, the mechanisms of failure of tumor-fighting T cells, and NK cells have been partially elucidated. The advances in the knowledge of different immune cell types in the TME, as well as the current and potential future use of this knowledge for designing more efficient immunotherapies are discussed in the following sections.

## 3. Immune Cell Constituents of the TME

### 3.1. Tumor-Associated Macrophages (TAMs)

Macrophages are phagocytic cells of the myeloid lineage that have a broad spectrum of functions including defense against invading pathogens, facilitating wound healing, and regulating tissue homeostasis [14].

Macrophages that infiltrate the microenvironment of solid tumors are called TAMs. During tumor progression, circulating monocytes in the peripheral blood are recruited to the tumor sites through chemokines secreted by tumors [15,16]. In addition, recent evidence has revealed that tissue-resident macrophages such as Kupffer cells, brain-resident macrophages, and alveolar macrophages contribute to the TAM population in different tumors [17,18].

Colony stimulating factor-1 (CSF-1) and monocyte chemoattractant protein-1 (CCL2) are the major chemokines for the recruitment of macrophages. Macrophages already present in the TME produce CCL2 and other chemokines, thereby generating a positive feedback loop for their recruitment. In a murine tumor graft model, depletion of CSF-1 led to reduced macrophage numbers and inhibited metastasis [19]. Similar to CCL2, there is a positive feedback loop for CSF-1-mediated bone marrow metastasis of breast cancer cells and recruitment of TAMs through the same chemokine axis [20].

In the TME, the polarization of macrophages into subtypes is controlled by soluble factors. Conventionally, macrophages are classified as M1 and M2 macrophages, having tumor inhibitory and tumor-promoting roles, respectively. M1 macrophages produce reactive oxygen species (ROS) and proinflammatory cytokines such as Interferon (IFN)-γ, Tumor necrosis factor (TNF)-α, Interleukin (IL)-2, and IL-1β, which play critical roles in killing tumor cells. Contrarily, M2 macrophages produce anti-inflammatory cytokines such as IL-10 and Tumor growth factor (TGF)-β, which promote tumor progression. Moreover, it is now clear that macrophages exist as an even more diverse spectrum of subtypes with high levels of plasticity [21].

At the tumor site, the contribution of M2 TAMs to tumor progression and immunosuppression of antitumor responses seems to overweigh their antitumoral responses. M2 TAMs contribute to tumor progression by inducing metastasis through the secretion of chemokines, by inducing angiogenesis, by promoting tumor growth through the release of growth factors and tumor cell invasion through the secretion of matrix metalloproteinases (MMPs), as well as by immunosuppression via cell–cell interactions and release of soluble factors [22]. In fact, the presence of TAMs correlates with poor prognosis in most tumors [23]. In this context, one important pro-tumorigenic axis is established via vascular endothelial growth factor (VEGF) secretion and induction of angiogenesis [24]. Macrophages contribute to ECM degradation via the release of MMPs, which enhances their invasiveness.

M2 TAMs further contribute to suppression of T cell antitumor responses by inducing exhaustion by high expression of ligands for exhaustion markers such as Programmed death-ligand 1 (PD-L1) and galectin (ligand for the T cell immunoglobulin and mucin domain-containing protein 3 TIM3). Exhaustion is a broad term describing a state of T cell dysfunction which arises in chronic viral infections and in cancer. At that state, T cells are defined by high expression of inhibitory receptors and a transcriptional signature different from effector and memory T cells and eventually by hypo-responsiveness.

At the molecular level, release of soluble factors including ROS and inhibitory cytokines, such as IL-10 and TGF-β, further dampen antitumor immunity. Thus, several layers of evidence have shown that TAMs have tumor-promoting, immunosuppressive roles in advanced solid tumors. Thus, TAMs are now considered to be one of the most important therapeutic targets [25].

### 3.2. Tumor-Associated Neutrophils (TANs)

For many cancer entities, the number of TANs correlates with poor prognosis [26,27,28], and the neutrophil-to-lymphocyte ratio (NLR) in the peripheral blood has currently been established as a prognostic marker for some cancer entities [29,30].

Neutrophils produce and release mediators that promote angiogenesis (e.g., VEGF, MMP-9, and Oncostatin M), modulate the microarchitecture of tumor tissues by releasing proteases, accelerate tumor cell proliferation, and induce genetic instability. In addition, TANs can facilitate immune evasion or can even be immunosuppressive by upregulation of PD-L1 [31,32,33], ROS production, or release of arginase. We have shown that ROS in the cell culture supernatant of activated human neutrophils leads to diminished migration and activation of T cells, for example, by inducing hypo-responsiveness, or can even induce T cell necroptosis [34,35]. One cellular target of ROS in T cells was identified as the actin-binding protein cofilin-1. Oxidized cofilin-1 loses its ability to induce actin dynamics and, consequently, leads to a stiffening of the actin cytoskeleton. Cofilin is also indirectly affected by arginase-1 (Arg-1) stored in tertiary granules of TANs. Arg-1 catalyzes the production of ornithine and urea from arginine. Thus, Arg-1 induces an extracellular arginine depletion which interferes with the signaling cascade, leading to cofilin activation. As a consequence, T cell proliferation and expression of IFN-γ is disturbed [36].

Another important tumor-promoting mechanism is NETosis, a process by which neutrophils expel net-like structures (neutrophil extracellular traps; NETs) made of chromosomal DNA in a suicidal manner into the extracellular space. NETs contain active proteases and oxidases [37]. The physiological function of NETs is to capture, opsonize, inactivate, and eliminate pathogens. A misdirected NETosis, for example, by sustained inflammation, increases the extracellular protease content, facilitating tumor cell migration and metastasis formation. Moreover, NET formation can awake dormant cancer cells, as was demonstrated in a lung cancer model [38]. NETs can also act as a protective hull on cancer cells against cytotoxic immune responses [39].

Contrarily, TANs may also mediate antitumor responses. Substances expressed by TANs, such as TNF-α or TNF-related apoptosis-inducing ligand (TRAIL), suppress tumor cell proliferation, induce tumor regression, or induce apoptosis in cancer cells [40,41,42,43]. From these different outcomes of TANs in the TME, it could be proposed that neutrophils are not a homogenous cell population, but that different neutrophil subgroups must exist. Based on mouse experiments, Fridlender et al. proposed that two groups of TANs existed, antitumor TANs (N1) and pro-tumor TANs (N2) [44]. N1 TANs have hyper-segmented nuclei and express TNF-α and CD95 (FAS). N2 TANs have circular nuclei and express ARG-1 and CCL2. While N1 TANs develop in the presence of type I interferons and are inhibited by TGF-β, N2 TANs develop in the presence of TGF-β and are inhibited by type I interferons. However, the classification is far from being complete. Another categorization of neutrophils reflects their physical properties in a density gradient centrifugation, and are defined as high-density neutrophils (HDNs) and low-density neutrophils (LDNs) [45,46]. HDNs develop from mature neutrophils, while LDNs develop from immature progenitor cells including N-MDSCs (see below) or from HDN in the presence of TGF-β. At the functional level, LDNs are comparable to N2 TANs and HDNs to N1 TANs (for more comprehensive descriptions of neutrophil subgroups, see a recent review by Jaillon et al.) [47].

### 3.3. Myeloid-Derived Suppressor Cells (MDSCs)

Myeloid cells originate from common myeloid progenitors which are multipotent immature myeloid cells (IMCs). These IMCs are believed to exist without immunosuppressive functions in healthy individuals. During myelopoiesis, multipotent progenitor cells differentiate into unipotent monocytes or neutrophils [48]. These myeloid cells represent the first line of defense during acute inflammatory conditions such as defense against invading pathogens [49]. Monocytes further migrate into tissues and differentiate into macrophages and DCs. However, during chronic inflammatory conditions such as cancer, IMC differentiation and maturation are impaired. A low and persistent level of exposure to inflammatory factors including cytokines such as IL-5, IL-6, IFN-γ, TNF-α, growth factors such as TGF-β, VEGF, GM-CSF, G-CSF, and various chemokines drive the differentiation of IMCs into MDSCs [50,51].

There are two major subsets of MDSCs characterized by their phenotypical and morphological characteristics as monocytic (M-MDCS) and neutrophilic (N-MDSC). Of these, a broad spectrum of M-MDCSs and N-MDSCs exist in the TME, which vary in the expression of their surface markers [52,53].

MDSCs contribute to tumor progression in multiple ways, including induction of angiogenesis, epithelial-to-mesenchymal transition (EMT), pre-metastatic niche formation, and immune evasion [54,55]. In fact, MDSCs have raised attention for their strong immunosuppressive functions. The mechanism of MDSC-induced immunosuppression occurs at multiple levels. These include production of ROS and reactive nitrogen species (RNS) [56,57] that cause T cell hypo-responsiveness and apoptosis [34,35,58], and the production and release of anti-inflammatory cytokines such as IL-10 and TGF-β, which suppress the effector immune cell functions [55,59]. MDSC-mediated T cell hypo-responsiveness is also induced metabolically through the deprivation of the amino acids arginine and cysteine, which are required for T cell proliferation and antitumor functions [60,61], as well as through the depletion of tryptophan by overexpression of indoleamine-pyrrole 2,3-dioxygenase (IDO), thereby leading to T cell anergy [62]. Moreover, MDSCs produce adenosine and result in inhibition of T cell activities [63].

At the cellular level, MDSCs hijack the immune checkpoint pathways, thereby acting as negative regulators of T cell and NK cell functions in the TME. In this context, upregulation of PD-L1 expression on MDSCs [64,65] was shown to induce T cell exhaustion [66]. PD-L1 expression on MDSCs from cancer patients has been reported in several studies [67,68], and PD-L1-positive MDSCs in peripheral blood have correlated with disease stage in lung cancer [69]. In contrast, the involvement of other checkpoint inhibitors, including TIM3 and cytotoxic T lymphocyte-associated protein 4 (CTLA-4), in MDSC-mediated T cell exhaustion has remained largely elusive.

### 3.4. Natural Killer Cells (NK Cells)

Natural killer (NK) cells are innate lymphoid cells that exert cytotoxicity similar to CD8+ T cells, albeit target cell recognition is different. They can exert cytotoxicity by direct contact with the target cells through the recognition of ligands by the activating NK cell receptor. The formation of cytolytic immune synapses and localized release of effector molecules, such as granzymes and perforin result in killing of target cells. NK cells also exert cytotoxicity by secretion of effector cytokines such as IFN-γ, TNF-α, IL-2, IL-12, IL-15, and IL-18. Thus, in a tumor environment, NK cells can control tumor growth by direct interactions and by controlling the function of innate and adaptive immune cells. However, unlike T cells and B cells, NK cells do not express clonotypic receptors and therefore are not dependent on antigen-specific activation [70]. NK cells have been historically divided into two groups, IFN-γ-producing CD56^high^CD16^low^ and cytotoxic (CD56^low^CD16^high^) NK cells. It is now clear that NK cells are heterogeneous, with variable expression of activating and inhibitory receptors [71]. NK cell cytotoxicity is mediated by a delicate regulation of inhibitory and activating signals that originate from their NK-inhibitory receptors (NK-IRs) and activating receptors (NK-ARs). During their development, licensing of NK cells through a process called “termed education” takes place, which prevents NK cells from attacking healthy cells. During this process, interaction of NK cells inhibitory tyrosine-based receptors with MHC-I helps to avoid killing healthy cells. NK-ARs recognize ligands derived from pathogens, cellular stress due to viral infections, or cellular growth factors [72].

Tumor cells often either lack or have low expression of MHC-I to escape destruction by CD8+ T cells. According to the “missing-self recognition” hypothesis, NK cells recognize cells that lack the MHC-I molecule, which disengages killer inhibitory receptors (KIRs), Ly49, and other MHC-I-specific inhibitory receptors. Thus, the evolution of this mechanism could be considered as a complement to the loss of T cell responses against MHC-I deficient tumor cells.

The presence of NK cells in the TME correlates with good prognosis in breast cancer [73], gastrointestinal stromal tumors [74], neuroblastoma [75], prostate cancer, lung cancer [76], and in several other tumors [77,78]. Despite this positive correlation in several solid tumors, many studies have revealed that tumor-infiltrating NK cells have altered expression of inhibitory and activating receptors, and overexpression of exhaustion markers [79,80], and eventually, impaired activities [81,82,83]. Multiple factors of the immunosuppressive solid tumor environment contribute to their impaired functions in the TME. One specific factor is the release of soluble NKG2D ligands by the tumor cells, which prevents interaction of NK cells with membrane-bound NKG2D ligands, thereby preventing their cytotoxic response [84]. Furthermore, in certain other tumors, there is either no or a negative correlation between the presence of NK cells and tumor progression [80,85]. Thus, a deeper understanding of NK cell functions in the TME, their phenotype, intercellular communication with other constituents of the TME, and the molecular constituents of the environment need to be further elucidated.

### 3.5. B Cells

Antibodies produced by B cells are the main constituents of the humoral immune system. The role of B cells in tumor progression and antitumor immunity is only beginning to be understood, partially due to their low numbers in the TME compared to T cells [86]. B cells that infiltrate into tumors, namely tumor-infiltrating B cells (TIBs), have been found in several studies and are considered an important prognostic factor in various tumors [87,88,89]. TIBs were found to be present in tumor-draining lymph nodes, and in tumor-associated tertiary lymphoid structures (TLS) [90,91]. Of note, TLS are lymphoid organs that develop ectopically in non-lymphoid organs under chronic inflammatory conditions including cancer. Well-developed TLS share the structural and functional characteristics with secondary lymphoid structures. These TLS can contain B cell follicles with actively replicating B cells surrounded by a T cell region. TLS can further help the maturation of memory B cells [92].

TIBs can have both tumor-promoting and antitumor functions. Accumulating evidence suggests that this largely depends on the subtype and activity of TIBs. Principally, TIBs exhibit antitumor functions by producing cytokines and antibodies, which induce antibody-dependent cellular cytotoxicity and phagocytosis and enhance antigen presentation by DCs. Importantly, B cells can also serve as antigen presenting cells (APCs), thereby shaping the T cell-mediated immunity in the TME [93].

Under normal conditions, DCs serve as professional APCs. However, during evolution of tumors, consistent interactions between T cells and tumor cells might lead to a reduction in DC numbers or activity, or could induce an immunosuppressive type of DCs (see below). Under these conditions, TIBs serving as APCs could help maintain antitumor T cell responses. In support of this assumption, in ovarian and liver cancer samples, surface markers associated with an APC phenotype were highly expressed on B cell populations [94,95]. TIBs were found to be concentrated in the margins of tumors and formed different sizes of tumor-associated tertiary lymphoid structures [94,96]. In these structures, antigen-specific interactions between T cells and B cells seem to be an important parameter in mounting an efficient antitumor immunity response since colocalization of CD20+ B cells and CD8+ T cells in breast cancer, in melanoma, and in ovarian cancer was found to be a positive marker [87,95,97]. In support of this, in a murine model, depletion of B cells by anti-CD20 antibody resulted in impaired antigen-specific activation and clonal expansion of CD4+ T cells, suggesting a critical role of B cells for T cell responses [98].

Moreover, in the TLS, B cells were reported to undergo clonal expansion, isotype switching, and tumor-specific antibody production [99,100]. The presence of a broad spectrum of antibodies recognizing tumor antigens (neoantigens) in the serum and in the TME supports the finding of clonal expansion and isotype switching of B cells against the tumors [101,102,103]. In fact, tumor-specific antibodies detected in the serum of several cancer patients are currently being considered for use as diagnostic biomarkers for early detection of cancer [101,102,104]. Overall, the presence of B cells in the TME has been positively correlated with better survival and lower relapse in colorectal carcinoma [105], lung cancer [106], breast cancer, ovarian cancer [107], melanoma [108], and in several other cancer types.

Notably, by contrast, the presence of TIBs has also been an indicator of a negative outcome in other human cancer types [109]. In this context, the tumor-promoting role of B cells has been attributed to a special type of B cells, namely regulatory B cells (Bregs). Bregs were shown to produce high levels of IL-10 and TGF-β which can downmodulate tumor-specific T cell- and NK cell-mediated responses [110,111], while increasing the frequency of Tregs in the TME [112]. Furthermore, Bregs were shown to have high PD-L1 expression and exert immunosuppression on CD8+ T cells through the PD-1/PD-L1 axis [113,114]. Considering the dual roles of TIBs in tumor immunity, the selective clearance of Bregs, promotion of TLS formation, as well as the targeted regulation of TIB-linked signaling pathways may become an effective means of TIB-based tumor immunotherapy [115].

### 3.6. Dendritic Cells (DCs)

DCs are central regulators of adaptive immune responses, providing antigen presentation and ligands for costimulatory receptors, as well as a suitable cytokine milieu for activation, differentiation, and effector functions of T cells (Figure 1). DCs were traditionally subdivided into conventional DCs (cDCs) and plasmacytoid DCs (pDCs). It is now evident that additional DC subtypes exist that express different markers. These include cDC1, cDC2, pDC, and monocyte-derived DC (moDC). Differentiation of these subtypes from a common DC progenitor is mediated by differential expression of transcription factors and can be discriminated by the expression of surface markers [116]. cDCs are normally in an immature state and require additional signals, such as inflammatory cytokines or “danger signals”, e.g., pathogen-associated molecular patterns (PAMPs), and danger-associated molecular patterns (DAMPs), for their maturation. Mature cDCs have a lower phagocytic capacity, increased levels of MHC-I/-II and costimulatory molecules, and enhanced cytokine production. Activated cDCs also have elevated levels of the lymph node-homing chemokine receptor CCR7 [117,118,119].

cDCs are associated with antitumor functions through endocytosis of dead tumor cells or debris, as well as through priming and activation of tumor-reactive T cells in tumor-draining lymph nodes by presentation of tumor (neo)antigens [120]. Importantly, the transcription factors IRF8, BATF3, and ID2 regulate the differentiation of cDC1 which express XRP, CD141, and Clec9a on their cell surface. cDC1 can preferentially cross-present exogenous antigens on MHC-I and activate CD8+ T cells [121,122]. In contrast, cDC2 express IRF4, ZEB2, RELB, and NOTCH2 master regulators CD11b, CD172A, as well as CD1c on their surface. cDC2 have a higher potential of MHC-II antigen presentation, and thereby increase CD4+ T cell activation [116].

Among others, the CD103+ cDC1 subtype has been shown to be the key DC subtype for priming of CD8+ T cells in tumor-draining lymph nodes [118]. cDC1 contribute to antitumor immunity by releasing the chemokines CXCL9 and CXCL10, which are essential for the recruitment of CXCR3+ effector T cells [123,124]. T cell effector functions also depend on cytokines such as IL-12 and type I interferons released from DCs.

Since DCs play a central role in orchestrating antitumor T cell responses in the TME, suppression of DCs poses a major bottleneck in antitumor responses. One major suppression of cDCs takes place by exclusion of cDC from the tumors through chemokine-mediated retention [125]. Another mechanism of downmodulation takes place via limited production of cytokines for cDC polarization and survival in the TME [126]. In this context, elevated IL-10 released by macrophages in the TME was shown to reduce production of IL-12, thereby leading to a diminished DC maturation [127]. Apart from these, the activity of cDCs in the TME can be further inhibited by metabolic challenges, such as competition for nutrients [128] or high concentrations of adenosine in the TME [129].

### 3.7. T Cells

T cells orchestrate the adaptive immune response against invading pathogens and transformed cells. In recent decades, the central role of T cells in antitumor immune responses has been extensively studied and is well recognized. In this context, the presence and distribution of tumor-infiltrating CD3+ and CD8+ T cells has been shown to be a positive prognostic factor in several cancer types. Immunoscoring based on CD3+ and CD8+ T cell presence and distribution in several tumors was correlated as a positive factor for survival in colon cancer, lung cancer, and in several others [4,130].

There are various T cell populations in the TME primarily located at the invasive margin and in the draining lymph nodes. The major populations are CD8+ cytotoxic T cells, CD4+ helper cells, and CD4+ regulatory T cells.

#### 3.7.1. CD8+ T Cells

CD8+ T cells are cytotoxic T cells that are activated upon engagement of the antigen-specific T cell receptor (TCR) by MHC-I presented cognate antigen on APCs in the presence of costimulatory signals. This is followed by massive clonal expansion and differentiation into different subtypes depending on the micromilieu, which is in part determined by secreted cytokines.

Initiation of a CD8+ T cell response against a tumor antigen in a tumor-draining lymphoid organ takes place through the presentation of tumor antigens by cDC1 [131]. The cross-presentation of exogenous tumor antigens in the context of MHC-I by DCs induces a CD8+ T cell response, leading to the elimination of tumors [132]. The interaction between CD70 and CD80/CD86 on APCs with CD27 and CD28 on CD8+ T cells is the key step in the priming of CD8+ T cells [133]. Furthermore, CD4+ T cells promote clonal expansion, and differentiation of CD8+ T cells into effector and memory subtypes. The CD4+ T cells help via CD40–CD40L interactions and promote proliferation of CD8+ T cells through IL-2 production [134].

Cytotoxic T cells (CTL) exert their functions primarily by releasing cytotoxins such as perforin and granzyme B to the target cell through cytolytic immune synapses or by inducing target cell apoptosis through FAS–FASL interactions [135]. Furthermore, activated CD8+ T cells express homing and chemokine receptors, thereby infiltrating infected or neoplastic tissue and typically producing high levels of IFN-γ and TNF-α to kill infected or tumorigenic cells. Importantly, recent evidence has clearly revealed that, similar to T helper lineages, CTL cell lineages, namely, Tc1, Tc2, Tc9, Tc17, and Tc22, develop in different environments. Each of these subtypes show differences in cytokine expression and in cytotoxic capacity [136]. In addition, different subtypes have been reported in different tumors, of which Tc1 was the most frequently found subtype [136].

The recruitment of CD8+ T cells to tumor draining lymph nodes is induced by the release of CCL9 and CCL10 from DCs, which bind to CXCR3 on CD8+ T cells [123]. CCL9 and CCL10 have also been reported to be produced by several tumors [137]. The presence of CD8+ T cells in breast cancer [138,139], colorectal cancer [140], ovarian cancer [141], melanoma [142], and several others has been associated with good prognosis.

During antigen clearance, the majority of effector cells die by apoptosis and a small proportion further differentiates into memory phenotypes. The memory subsets have heterogenous phenotypes which can be categorized based on the expression of the respective surface markers (Table 1) as well as other receptors such as killer cell lectin-like receptor G1 (KLRG1), CD27, IL7R, and CXCR3. CD8+ T cells are classically subdivided into naïve, effector, and distinct memory phenotypes based on their surface expression of CCR7, CD45, and CD62L expression [143]. Single-cell sequencing and cytometry by time of flight (CyTOF) techniques advanced this narrow understanding of classically identified subtypes into a broad subtype of CD8+ T cells with varying functions [144] (Table 1). It is known that effector memory (T_EM_) and central memory (T_CM_) cells circulate into the bloodstream and in secondary lymphoid organs. T_EM_ cells have limited expansion capacity and are in a terminally differentiated state. Contrarily, memory stem cells (T_SCM_) have the capacity to rapidly respond to antigens and effector molecules and can generate memory and effector cells. The differentiation from memory to terminally differentiated cells is regulated by various transcription factors for the memory phenotype; these are Eomes, TCF-1, Bcl6, and ld3, and for the differentiated phenotype, BLIMP, STAT4, ld2, and T-bet [145].

The importance of the characterization of T memory cell subsets can be best understood from immunotherapeutic studies that show a positive correlation between the presence of CD8+ memory T cells and antigen clearance. In this context, the TILs that responded to checkpoint blockade therapy were mostly TCF-1 positive memory T cells [146,147,148].

Importantly, the presence of large amounts of tumor-infiltrating CD8+ T cells in the TME of progressed tumors suggested a dysfunctional state in T cells. These cells are highly exhausted cells with an altered transcriptional regulation, cytokine profile, and surface marker expression. Such dysfunctional or exhausted CD8+ T cells express high levels of PD-1, lymphocyte-activation-gene 3 (LAG3), TIM3, T cell immunoreceptor with Ig and ITIM (TIGIT), and CTLA-4 [145,150]. The exhausted cells do not respond to TCR stimulation, have a reduced proinflammatory cytokine production capacity, and a reduced ability to kill tumors. Therefore, significant efforts have been made to reactivate exhausted CD8+ T cells.

Apart from these well-known coinhibitory molecules such as PD-L, CTLA-4, several other markers have been identified [151] (Table 2). However, the contribution of the other markers to the fate of T cells remains largely elusive.

Additionally, it should be noted that the immunosuppressive TME poses several other direct and indirect challenges to prevent various steps of tumor elimination by CD8+ T cells. These include the failure of cross-priming by DCs, thereby preventing generation of primed-CD8+ T cells and inhibition of tumor-infiltrating CD8+ T cell functions by induction of metabolic changes and oxidative stress.

#### 3.7.2. CD4+ T Helper Cells

CD4+ T cells are polyfunctional, versatile adaptive immune cells with a differentiation capacity to several functional subtypes depending on the signals. Their activation and differentiation requires antigenic stimulation, costimulatory signals and a specific cytokine milieu [153]. CD4+ T cells have a central role in antitumor immunity since they regulate the functions of the majority of the tumor-infiltrating leukocytes, including CD8+ T cells, NK cells, macrophages, and DCs.

CD4+ T cells contribute to the antitumor response directly by eliminating tumors through cytolytic mechanisms mediated by the release of IFN-γ and TNF-α, and indirectly by modulating the presence and activity of immune cell infiltrates in the TME [154,155] (Figure 2). Of note, CD4+ T cell cytotoxicity through FAS–FASL interactions as well as granzyme B (GrzB) and perforin release during viral infections is a known phenomenon [156]. However, whether a similar way of killing by CD4+ T cells take place in tumors remains to be elucidated.

In secondary lymphoid organs, CD4+ T cells contribute to the differentiation of B cells and CD8+ T cells. CD4+ T cells can help the CD8+ T cell-mediated antitumor response in multiple ways. First, interaction of CD40L on CD4+ T cells with CD40 on DCs is essential for CD8+ T cell priming [133]. CD40–CD40L signaling leads to optimal antigen presentation by DCs through the induction of expression of costimulatory molecules and cytokines. Maturation and expansion of memory CD8+ T cells are also regulated by CD4+ T cells through the licensing of DCs and through the direct release of cytokines such as IL-15 during priming (Figure 2) [157,158,159]. CD4+ T cell help for B cells is also mediated by CD40L and CD40 interactions. Guy et al. showed that CD4+ T cell help leads to the generation of isotype-switched antitumor antibodies. In the absence of CD4+ T cells, agonistic anti-CD40 administration led to antibody isotype switching and lower metastasis rates. However, this did not prevent the growth of subcutaneous tumors, confirming additional roles of CD4+ T cells for controlling the growth of tumors [160].

In recent years, the antitumor role of CD4+ T cells has gained greater interest since several melanoma (neo)antigens were found to be recognized by CD4+ T cells both in murine models as well as in human melanomas [161,162]. In support of these findings, two studies further showed that mRNA vaccination with (neo)antigens mounted a strong CD4+ T cell response [163,164]. These studies were followed up by several others which consolidated the understanding of CD4+ T cell help for CD8+ T cell-mediated antitumor immunity [165,166], and which identified unique populations of CD4+ T cells that have direct cytolytic activities [167].

CD4+ T cells comprise several subtypes whose development and effector function are well-described. However, the contribution of individual subtypes to antitumor functions or tumor progression is less well-understood. Among the CD4+ T cell subtypes, namely, Th1, Th2, Th17, Th9 and Th22 cells, Th1 is the most abundant and potent subgroup in various cancer types. Th1 cells produce high levels of IFN-γ. In an IFN-γ-dependent manner, they can activate cytotoxic functions of DCs [168]. This can further increase the presentation of different tumor-specific antigens by DCs [169]. Furthermore, Th1 cells produce chemokines that enhance the priming and expansion of CD8+ T cells, and the recruitment of NK cells and M1 macrophages [170]. In fact, Th1 promotion by IL-12 mRNA injection has strongly increased IFN-γ levels, cytotoxic T cell numbers, and antitumor responses in murine models [171].

In contrast, the antitumor effects of Th2 cells are rather contradictory. Secretion of IL-4 by Th2 cells can exert direct antitumor effects [172] and increase the recruitment and the maturation of macrophages and eosinophils [173]. However, considering this dual role, the mostly tumor-promoting role of M2 macrophages and eosinophils, and the Th2-mediated immunity could be context-dependent. Similar to Th2 cells, Th17 and Th9 cells have been identified in several cancer types [174,175,176]. However, their roles have been controversial as they were reported to have both antitumor and tumor-promoting functions [177]. Th17 cells require IL-6, IL-1β, TGF-β, and IL-21 for their differentiation and produce pro-inflammatory IL-17 cytokines as well as IL-23, IL-21, and GM-CSF. The controversial roles of Th17 cells in cancer can partially be explained by the high plasticity of Th17 cells, their potential effects on TANs, and their potential differentiation into other cell types [178]. In the TME of different cancer types, the presence and concentration of various cytokines, including TGF-β, IL-4 and IFN-γ and of other molecules, such as ROS, can change the phenotype and function of Th17 cells [179,180].

#### 3.7.3. Regulatory T Cells

Regulatory T cells (Tregs) are a major subset of CD4+ T cells, with tolerogenic and immunosuppressive functions. Tregs are characterized by high surface expression of CD25, low expression of CD127, and expression of the transcription factor FOXP3, the latter being the master regulator of the subset [181]. Decades of research on Tregs have now shown that these can be categorized into two subsets based on their origin. Tregs that are produced in the thymus make up the majority of Tregs in the body and are called naturally occurring thymic Tregs (tTregs). The development of thymic Tregs is dependent on high-avidity interactions with self-antigens and IL-2 signaling in the thymus. Tregs that develop in the periphery from naïve CD4+ T cells upon exposure to antigen stimulation under certain cytokine milieu are called peripheral Tregs (pTregs) [182,183]. Both populations are recruited in the TME and pTregs can further be induced in an immunosuppressive TME [184].

Tregs are one of the major cell types inducing immunosuppression in various tumors [185]. Treg-dependent suppression of antitumor immunity is associated with different functions at the cellular and molecular level. The cellular level of inhibition is primarily exerted by cell–cell interactions via the CTLA-4 axis. In this context, Tregs express high levels of CTLA-4 that bind to CD80/CD86 costimulatory molecules on DCs, thereby blocking their availability for binding of the costimulatory receptor CD28. This results in diminished T cell activation and proliferation [186]. Furthermore, through the LAG-3 MHC-II axis, Tregs can prevent the maturation of DCs, thereby impairing costimulation and activation of tumor-reactive T cells [187].

At the molecular level, Treg-mediated immunosuppression is exerted primarily by the release of anti-inflammatory cytokines, competition with T cells for metabolites, and increasing extracellular adenosine levels. The release of anti-inflammatory cytokines, such as TGF-β and IL-10, exerts strong and multiple inhibitory roles in antitumor immunity [188,189]. Tregs also compete with other T cells for IL-2 and the limited IL-2 levels can diminish T cell proliferation [190]. Another molecular control of T cell functions by Tregs takes place by the conversion of extracellular ATP to adenosine [191]. Under inflammatory conditions, such as cancer, high levels of ATPs are released in the extracellular space. Extracellular ATP is converted to ADP by CD39 and dephosphorylated to adenosine by CD73, which are highly expressed on the Treg surface [191,192]. Extracellular adenosine then binds to the A2A receptor on T cells, resulting in increased intracellular cAMP levels and inhibition of T cell function [193,194]. Thus, the presence of Tregs in the TME of most cancers is correlated with poor prognosis, making Tregs a major target for immunotherapies.

## 4. Molecular Constituents of an Immunosuppressive TME

Soluble factors that block T cell metabolism, activation, and effector functions and increase immune tolerance contribute to immunosuppression in the TME. In this context, the immunoinhibitory cytokines, such as TGF-β and IL-10, the pro-oxidative milieu [58,195,196,197,198], and metabolic factors are the central micromilieu elements responsible for immunosuppression in the TME.

### 4.1. Anti-Inflammatory Cytokines

Anti-inflammatory cytokines that are either secreted by tumor cells or by tumor-hijacked MDSCs, M2 TAMs, N2 TANs, and Tregs further limit antitumor immunity. In this context, TGF-β and IL-10 are the best-known mediators of immunosuppression. The tumor-promoting roles of TGF-β during tumor progression are associated with different functions including ECM remodeling, EMT transition, and formation of an immunosuppressive TME. TGF-β can suppress the activity of DCs [199], and can directly inhibit T cells and NK cells at several states [200] while promoting Tregs and MDSCs [201]. TGF-β is associated with immunosuppression in many cancer types. In this context, recent studies have revealed that TGF-β is important for the exclusion of T cells from tumors, leading to immunologically cold tumors [202,203], and that targeting of TGF-β can sensitize tumors for combination therapies [204]. Similarly, IL-10 suppresses DCs and T cell function at several states. Targeting of IL-10 has shown promising results in preclinical studies [205]. However, since the role of IL-10 is context-dependent and several sources of IL-10 exist, systemic targeting of IL-10 holds potential risks. Therefore, additional molecular insights into the role of IL-10 for tissue homeostasis both within and outside of the tumor are required [206].

### 4.2. Reactive Oxygen Species (ROS)

Apart from the cytokines, a central regulator of the TME is the redox micromilieu. Tumors are known entities with a special redox homeostasis and typically possess a pro-oxidative micromilieu [207,208,209,210]. In malignant cells, ROS result from increased basal metabolic activity, mitochondrial dysfunction, uncontrolled growth factor or cytokine signaling, and oncogene activity, as well as from enhanced activity of certain ROS-producing enzymes such as NADPH oxidases (NOXes). Apart from intracellular production in tumor cells, ROS derive from several extracellular sources in various tumors. Stromal cells, TAMs, TANs, MDSCs, and endothelial cells are the cellular sources of ROS that make up the pro-oxidative milieu of solid tumors [211] (reviewed in [212,213]). Furthermore, many cells in the tumor environment undergo cell death via apoptosis or necrosis due to low access to O_2_, nutrients, or specific killing by tumor-specific lymphocytes, which eventually leads to the release of H_2_O_2_ and other ROS into the surroundings (reviewed in [214]). Intriguingly, the majority of chemotherapeutic reagents function directly or indirectly by inducing ROS. While high levels of ROS have been shown to induce apoptosis, low levels and spatiotemporally controlled ROS generation can induce tumorigenesis [215,216].

At high concentrations, ROS are detrimental due to their irreversible effects on protein thiols (hyperoxidation), on lipids, and on DNA. However, low levels of ROS are known to regulate key signaling events and functions of the cells. Intriguingly, tumor cells overexpress their antioxidant systems to cope with the high ROS levels [217,218]. Conversely, functions of T cells, NK cells, and potentially B cells, which have a lower antioxidant capacity, are strongly downmodulated by high ROS levels in the TME. In this context, we have previously shown that proteins modulating actin cytoskeletal dynamics (cofilin and L-plastin) are highly oxidized in T cells under pro-oxidative conditions [34,35,219,220]. Since these proteins regulate various T cell functions, including T cell migration, invasion, and cytotoxicity, the oxidation of such proteins can partly explain the loss of T cell functions in a pro-oxidative TME. Similarly, the NK cell response has been reported to be downmodulated under pro-oxidative conditions [221]. However, an elaborate analysis of global oxidation is required for a better understanding of redox-regulated T cell, B cell, and NK cell functions and their respective failures in a pro-oxidative TME.

### 4.3. Metabolites

Another important molecular control of antitumor immunity is exerted via metabolites. Tumor cells usually switch their metabolism to glycolysis under hypoxic conditions and even under aerobic conditions. Similarly, activated T cells, specifically effector T cells, heavily rely on glycolysis. Thus, tumor cells and TILs compete for glucose as an energy source and as the precursor for biosynthesis of other molecular precursors [25,222]. Apart from this, immune cell functions are inhibited by deprivation of the amino acids tryptophan, cysteine, and arginine, as well as by elevated levels of free adenosine in the TME. Principally, depletion of arginine is mediated by the release of arginase into the TME. TANs are a well-known source of arginase, whose release was shown to downmodulate T cell functions [223]. Similarly, MDSCs also release arginase in the TME, thereby contributing to arginine depletion and T cell hyperresponsiveness [224]. Deprivation of another essential amino acid, tryptophan, is mediated by high expression and extracellular levels of IDO in the TME [225]. IDO1 is associated with a poor prognosis in solid tumors [226] and IDO1 inhibition in tumors was shown to enhance T cell responses in murine models [227]. As mentioned above, free adenosine in the TME binds to the A2AR on T cells, which results in the inhibition of antitumor immune responses [63,194].

## 5. Immunotherapeutic Strategies against Solid Tumors

To bypass the immunosuppression in the TME, several immunotherapies, particularly T cell-based immunotherapies, are being developed. Specifically, T cell-based therapies (adoptive T cell therapy) including ex vivo expanded tumor-infiltrating lymphocytes (TIL), engineered T cell receptor (TCR) as well as chimeric antigen receptor (CAR) T cells have opened a new era in cancer therapy [228,229,230]. In addition to these, DC vaccination therapy as well as engineering of NK cells, macrophages, and targeting of soluble factors and metabolites in the TME are other types of immunotherapies (Figure 3). Furthermore, antibody therapies aiming at checkpoint blockade as well as antibody-dependent cellular cytotoxicity (ADCC) are successful strategies that are currently being evaluated in preclinical models and in clinical trials. Below, we focus on T cell-based immunotherapies.

### 5.1. The Portfolio of T Cell-Based Immunotherapies

#### 5.1.1. TIL Therapy

The TIL therapy was pioneered in the 1980s by Rosenberg and colleagues, where the TIL were grown from murine tumors ex vivo in the presence of IL-2 and were shown to reduce the metastasis in different cancer models [231]. Currently, TIL therapy includes ex vivo expansion of TILs from resected tumors followed by adoptive transfer to the patients after lymphodepletion. Briefly, TILs are cultured in the presence of IL-2. Thereafter, TILs are exposed to rapid expansion protocols (REPs), where restimulation is performed via monoclonal anti-CD3 antibodies in the presence of allogeneic irradiated peripheral mononuclear cells and IL-2 [232,233]. TILs have been successfully generated from various tumors including melanoma, cervical cancer, renal cell cancer, breast cancer, and non-small cell lung cancer [234]. Nonetheless, the TILs have shown consistent antitumor functions only against melanoma, while their reactivity against other cancer types remained poor. One major reason for this is the different (neo)antigen availability, and the presence of different clones of TILs in melanoma and lower numbers of clones in other solid tumors. Furthermore, different solid tumors pose different challenges depending on the cellular and molecular contexture of the TME. Of note, most of the previously published studies have focused on a rapid expansion of TILs, even in the absence of co-stimulation. It has now become evident that T cells require not only IL-2 supplementation, but also stimulatory and co-stimulatory signals. In addition, further characterization of the TIL subsets (Th or Tc subtypes), the memory state, as well as the exhaustion state need to be considered. Furthermore, in the ex vivo, expanded TILs-specific depletion of Tregs would be a considerable improvement as their ex vivo expansion would counteract the potential antitumor activity of TILs. Perhaps, for a more efficient TIL therapy in future, the objective could be to achieve a memory phenotype and a diminished exhaustion state of TILs.

#### 5.1.2. TCR Therapy

The other TCR-based adoptive T cell therapy includes engineering of autologous T cells with genes encoding for TCRα and β that recognize specific tumor antigens. The expression of the genes was previously performed via retroviral or lentiviral transduction. The CRISPR/Cas9-based technology or “sleeping beauty” technology represent other means of genetic engineering currently being applied [235]. Using these TCR-based therapeutic strategies, TCR-modified T cells specific for MART-1, MAGE, NY-ESO1, or specific CEA tumor antigens have been generated and tested in clinical trials [236,237,238]. TCR-based adoptive T cell therapies have the advantage of generating several tumor antigen-specific T cells with a defined background. However, tumors can easily escape this therapy by reducing the expression of their MHCs. Another major drawback of this therapy is the HLA restriction, which prevents usage in patients with different HLA haplotypes.

#### 5.1.3. CAR T Cell Therapy

Among the immunotherapeutic strategies, CAR T cells represent a very promising concept for combating cancer. They are engineered to recognize a specific antigen, for example, neoantigens, on tumor cells and generate a tumor-specific cytotoxicity. Briefly, the transduced CAR is composed of (i) an ectodomain derived from a single chain variable fragment of an antibody that recognizes a (neo)antigen, (ii) the transmembrane domain, and (iii) an endodomain with intracellular signaling domains derived from the CD3 ζ chain and co-stimulatory molecules. This structure allows a specific recognition of a (neo)antigen on tumors, resulting in T cell-mediated tumor cytotoxicity in an MHC-independent manner. CD19-specific CAR T cells have shown great effects against B cell hematologic malignancies, which have achieved up to 90 % remission rates in clinical trials [239]. CAR T cell therapy in solid tumors has, so far, been unsatisfactory due to serious side effects and/or lack of therapeutic responses [240]. The failure of CAR T cells against solid tumors is multifactorial and is highly likely due to a combination of the following suppressive components of the TME: diminished tumor infiltration, exhaustion, and the presence of inhibitory cell types such as MDSCs, TAMs, and TANs. At the molecular level, the presence of anti-inflammatory cytokines, adenosine, and ROS potentially further contributes to their failure in this environment. Therefore, novel strategies in combination with CAR T cells are required. In fact, improvement of CAR T cell infiltration into tumors by co-expression of CCR4, and CCR2 chemokine receptors have enhanced the antitumor responses in preclinical models [241,242]. Similarly, CAR T cells modified with shRNAs targeting exhaustion markers have shown better efficacy in murine models [243,244]. Furthermore, CAR T cells simultaneously expressing IL-12 [245] or IL-7 [246] have shown improved efficacy.

Apart from CAR-intrinsic improvements, a combination of CAR therapy with other therapies have shown enhanced efficacy. In this regard, the combination of CAR T cell therapy with anti-PD-1 checkpoint inhibitor therapy showed enhanced efficacy in preclinical models and also in clinical trials [247]. Along the same line, several attempts are being made to combine CAR T cell therapy with virotherapy for a more efficient antitumor response in solid tumors [248,249,250]. The latter approach holds promise to specifically lyse tumor cells as well as to target suppressive constituents of the TME specifically by carrying other factors, such as a bi-specific T cell engager (BITE) or cytokines [236]. However, the tropism and lytic capacity of such viruses require considerable advancements for enhanced efficacy and minimal toxicity. Despite these improvements, the global and strong empowerment of CAR T cell functions remains to be elucidated. It seems that improving the tumor-infiltration capacity, diminishing exhaustion, or expression of cytokines are not sufficient. Potentially, improvement of their survival, proliferation, and effector functions also require a resistance to the micromilieu.

## 6. Concluding Remarks

In the era of tumor immunotherapy, varying levels of success of different immunotherapies have been achieved in several tumor types. Specifically, cell-based therapies such as CAR T cell therapies against B cell hematological malignancies have shown strong responses. Yet, the failure of therapies against the majority of solid tumors overweighs the successes until now. Nonetheless, it is exciting to note that even the failure of these cell-based, antibody-directed, or micromilieu-targeting novel immunotherapies is accompanied by new lessons, as well. These failures have driven curiosity to understand the molecular mechanisms of immunosuppression and to develop strategies for overcoming the barriers of the immunosuppressive TME.

One fundamental reason for the failure of immunotherapies in solid tumors is the existence of multiple immunosuppressive elements in the TME. In addition, there is a complex interplay among different molecular and cellular constituents of the TME. For instance, TGF-β signaling induces the activation of ROS-producing enzymes, namely NOXes, thereby contributing to a pro-oxidative environment. Along the same line, ROS can regulate certain metabolic enzymes such as GAPDH, and thus could potentially influence functions and memory subsets of tumor-infiltrating T cells [251]. Tumor cell with high antioxidant capacity can limit the influence of ROS, while T cells with lower antioxidant capacity may fail to do so. It should be noted that hypoxic conditions can also lead to an elevated ROS production, further contributing to the pro-oxidative micromilieu [252]. Taken together, a better understanding of the molecules, their producers, and their mechanism of inhibition on tumor-reacting immune cells is required for successful immunotherapies. In this context, the pro-oxidative micromilieu of tumors and redox regulation of immune cell functions as well as tumor progression requires special attention.

The immune cell constituents of the TME in various cancer types differ in their abundance, cell types, subsets, and in their activation and exhaustion states. Similarly, different TME also greatly vary in terms of their non-immune cell context, which can further influence the behavior of the immune cell constituents. Evidently, an immunosuppressive TME receives a synergistic contribution from the molecular and cellular constituents of the TME. The majority of the MDSCs, TAMs, TANs, and Tregs contribute to the pro-oxidative, acidic, adenosine-rich, and anti-inflammatory cytokine-rich TME. Under these conditions, most T cell-based therapies including CAR T cell therapies fail against solid tumors. Therefore, engineering tumor-reactive T cells resistant to specific immunosuppressive TME components and therapies targeting the same or other immunosuppressive components can be combined. This also requires patient- and tumor-specific identification of major contributors to the immunosuppression for the targeting and specific empowerment of T cells. Moreover, the dynamic response of the TME to such therapies needs to be monitored at the molecular level during treatment, possibly also by radiogenomic approaches [253]. This will allow for the redefining of the modality of immunotherapy depending on potential resistance mechanisms acquired by the TME in response to treatment. Taken together, the design of combinatorial immunotherapies requires consideration of the dynamics of the TME. Thus, a deeper understanding of the cellular and molecular dynamics within the TME is required for designing combinatorial immunotherapies that allow a more successful treatment of solid tumors in the future.

## Figures and Tables

**Figure 1 ijms-22-05736-f001:**
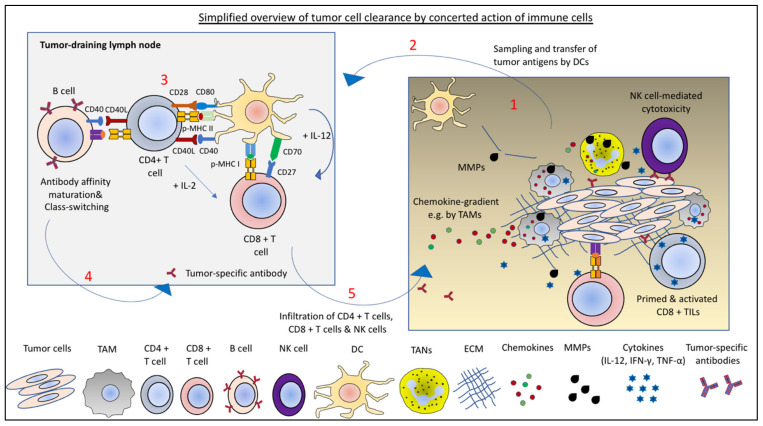
Tumor cell clearance by the concerted action of immune cells. (**1**) Sampling of tumor antigens by DCs and (**2**) antigen-presentation in tumor-draining lymph nodes activates CD4+ T cells and CD8+ T cells. (**3**) Priming and activation of CD8+ T cells further requires licensing of DCs by CD4+ T cells and the cytokines produced by activated CD4+ T cells. (**4**) Activated T cells help B cell maturation, antibody class-switching, and production of tumor-specific antibodies. NK-cell mediated cytotoxicity can take advantage of the tumor-specific antibodies. (**5**) Activated, primed cells infiltrate the tumors. The tumor infiltration of activated immune cells require chemokine matching as well as ECM degradation and remodeling by MMPs.

**Figure 2 ijms-22-05736-f002:**
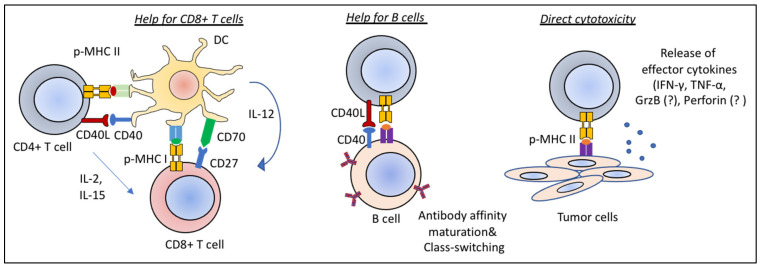
Antitumor functions of CD4+ T cells. CD4+ T cells exert antitumor functions by helping CD8+ T cells and B cells, and through direct cytotoxicity. Licensing of DCs by CD40–CD40L interactions is critical for priming of CD8+ T cells (left). The CD40–CD40L axis also orchestrates CD4+ T cell help to B cells leading to antibody affinity maturation and isotype-switching (middle). CD4+ T cells are also able to perform direct cytolytic functions whereby IFN-γ and TNF- α play critical roles. A direct cytotoxicity of CD4+ T cells against tumors by release of Granzyme B and Perforin, a known phenomenon during viral infections, likely also takes place in tumors (right).

**Figure 3 ijms-22-05736-f003:**
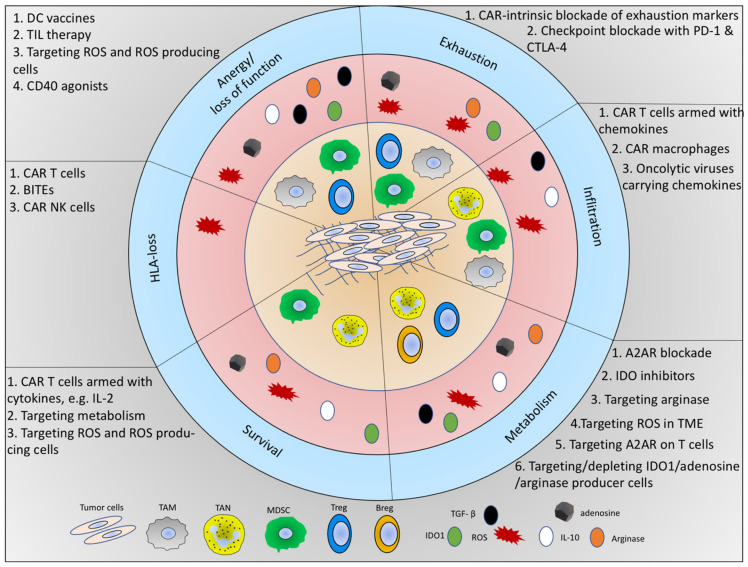
Schematic overview of immunosuppressive components of the TME, the immunosuppression on T cells, and the current immunotherapeutic strategies to overcome immunosuppression. The inner circle (light orange) shows the tumor cells and immunosuppressive immune cell types in the TME. The middle circle (red) indicates the immunosuppressive molecules. The outer circle (light blue) indicates the mechanism of inhibition on tumor-fighting immune cells (CD4+ T cells, CD8+ T cells, and NK cells). For each of these major mechanisms, the cells and the soluble factors released by the respective cells are shown in six slices. In the grey area of each slice, the current immunotherapeutic strategies are listed.

**Table 1 ijms-22-05736-t001:** T cell subsets and their antitumor capacity. The surface markers that discriminate the different subsets, as well as the proliferation capacity and the effector functions of each population are given [143,144,145,149].

T Cell (Memory) Subset	Surface Markers							Antitumor Functions		
	CD45RA	CD45RO	CD28	CD27	CD57	CD62L	CD127	CCR7 (CD197)	Proliferation Potential	Effector Functions
Naïve (T_NA_)	+	−	+	+	−	+	+	+	+/−	−
Central memor (T_CM_)	−	+	+	+	−	+	+	+	+/−	+
Effector memory (T_EM_)	−	+	−	+/−	+	−	+/−	−	−	+
Resident memory (T_RM_)	−	+	+/−	+	−	−	+	−	?	+
Memory stem cells (T_SCM_)	+	+	+	+	−	+	+	+	+	+
Effector memory RA cells (T_EMRA_)	+	+	−	−	+	−	−	−	−	+
Terminally differentiated effector cells (T_EF_)	+	−	−	−	+	−	−	−	−	++

**Table 2 ijms-22-05736-t002:** Markers of T cell exhaustion and their ligands on different cell types [145,150,151,152].

Exhaustion Marker Expressed on T Cell	Ligand	Cells Expressing the Ligand	Influence on Antitumor Functions of T Cells
PD-1	PD-L1	Tumor cells, DCs, TAMs, Bregs	Inhibitory
CTLA-4	CD80/CD86	DCs, B cells, TAMs, Treg	Inhibitory
TIM3	Galectin 9	APC	Both inhibitory and activating, but mostly inhibitory
LAG3	MHC-I or MHC-II–Peptide complex	APC, tumor cells	Both inhibitory and activating, but mostly inhibitory
TIGIT	CD155	APC	Inhibitory
Vista	?	APC	Largely remains to be elucidated
CD96	CD112	APC, tumor cells	Largely remains to be elucidated
BTLA-4	HVEM	T cells, Tregs, APCs, tumor cells	Largely remains to be elucidated
CD160	HVEM	T cells, Tregs, APCs, tumor cells	Largely remains to be elucidated

## Data Availability

Not applicable.

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
