# Peer review of "Hijacked Immune Cells in the Tumor Microenvironment: Molecular Mechanisms of Immunosuppression and Cues to Improve T Cell-Based Immunotherapy of Solid Tumors"

_ijms, 2021, doi:10.3390/ijms22115736_

Round 1

Reviewer 1 Report

The authors proposed to review the molecular mechanisms of immunosuppression and cues to improve T cell based IMT in solid tumors. The review is thorough and timely for the field. I have few suggestions to improve it.

  1. It will be greatly helpful, if the authors can include a section to discuss the emerging biomarkers (from clinical trials or retrospective analysis) to predict the IMT response, and their link with biological mechanisms studies.
  2. Besides the TME, the host immune environment and other factors can affect the TME and IMT response, can authors also provide some review on this part?
  3. Besides the biological mechanisms study, imaging (radiomics, deep learning) and radiogenomics studies have been proposed to predict the response to IMR (my group has published a recent paper https://doi.org/10.1016/j.semcancer.2020.12.005), the authors may want to discuss this point. 

Reviewer 2 Report

Samstag et al contributed a well-organized, well-written review paper focusing TME in solid tumors. The authors discussed a wide range of components in TME that shaped immune evasion, including immune cells and other factors. They proposed a variety of corresponding strategies to tackle the immunosuppression in TME. The manuscript is acceptable for publication in this journal in the present form.
